# Low incidence of antibiotic-resistant bacteria in south-east Sweden: An epidemiologic study on 9268 cases of bloodstream infection

Martin Holmbom[1,2], Vidar Möller[1], Lennart E. Nilsson[3], Christian G. Giske[4,5], Mamun-Ur Rashid[1,4], Mats Fredrikson[6,7], Anita Hällgren[1], Håkan Hanberger[1]*, Åse Östholm Balkhed[1]

1 Department of Infectious Diseases in Östergötland and Department of Biomedical and Clinical Sciences, Linköping University, Linköping, Sweden, 2 Department of Urology in Östergötland and Department of Biomedical and Clinical Sciences, Linköping University, Linköping, Sweden, 3 Department of Clinical Microbiology and Department of Biomedical and Clinical Sciences, Linköping University, Linköping, Sweden, 4 Department of Laboratory Medicine, Karolinska Institute, Stockholm, Sweden, 5 Clinical Microbiology, Karolinska University Hospital, Stockholm, Sweden, 6 Department of Biomedical and Clinical Sciences, Occupational and Environmental Medicine, Faculty of Medicine and Health Sciences, Linköping University, Linköping, Sweden, 7 Forum Östergötland, Faculty of Medicine and Health Sciences, Linköping University, Linköping, Sweden

☯ These authors contributed equally to this work.
* hakan.hanberger@liu.se

**Data Availability Statement:** All relevant data are within the manuscript and its Supporting Information files.

## Abstract

### Objectives

The aim of this study was to investigate the epidemiology of bloodstream infections (BSI) in a Swedish setting, with focus on risk factors for BSI-associated mortality.

### Methods

A 9-year (2008–2016) retrospective cohort study from electronic records of episodes of bacteremia amongst hospitalized patients in the county of Östergötland, Sweden was conducted. Data on episodes of BSI including microorganisms, antibiotic susceptibility, gender, age, hospital admissions, comorbidity, mortality and aggregated antimicrobial consumption (DDD /1,000 inhabitants/day) were collected and analyzed. Multidrug resistance (MDR) was defined as resistance to at least three groups of antibiotics. MDR bacteria and MRSA, ESBL-producing Enterobacteriaceae, vancomycin-resistant enterococci not fulfilling the MDR criteria were all defined as antimicrobial-resistant (AMR) bacteria and included in the statistical analysis of risk factors for mortality

### Results

In all, 9,268 cases of BSI were found. The overall 30-day all-cause mortality in the group of patients with BSI was 13%. The incidence of BSI and associated 30-day all-cause mortality per 100,000 hospital admissions increased by 66% and 17% respectively during the nine-year study period. The most common species were *Escherichia coli*, *Staphylococcus aureus*, *Klebsiella pneumoniae*, *Streptococcus pneumoniae* and *Enterococcus faecalis*.

**Funding:** The authors received no specific funding for this work.

**Competing interests:** The authors have declared that no competing interests exist.

Independent risk factors for 30-day mortality were age (RR: 1.02 (CI: 1.02–1.03)) and 1, 2 or ≥3 comorbidities RR: 2.06 (CI: 1.68–2.52), 2.79 (CI: 2.27–3.42) and 2.82 (CI: 2.31–3.45) respectively. Almost 3% (n = 245) of all BSIs were caused by AMR bacteria increasing from 12 to 47 per 100,000 hospital admissions 2008–2016 (p = 0.01), but this was not associated with a corresponding increase in mortality risk (RR: 0.89 (CI: 0.81–0.97)).

## Conclusion

Comorbidity was the predominant risk factor for 30-day all-cause mortality associated with BSI in this study. The burden of AMR was low and not associated with increased mortality. Patients with BSIs caused by AMR bacteria (MDR, MRSA, ESBL and VRE) were younger, had fewer comorbidities, and the 30-day all-cause mortality was lower in this group.

## Introduction

The high burden of bloodstream infection (BSI) and increasing prevalence of BSI caused by antimicrobial-resistant (AMR) and multidrug-resistant (MDR) bacteria is a serious threat to global public health [1–6]. In Europe, BSIs caused by *Escherichia coli* resistant to third-generation cephalosporins (ESBL phenotype) have increased to a EU/EEA population-weighted mean percentage of 15% in 2017, whereas the corresponding mean for methicillin-resistant *Staphylococcus aureus* (MRSA) has decreased from 20% in 2014 to 17% in 2017 [7]. MRSA causes only 1.2% of all *S. aureus* BSIs in Sweden and has not increased significantly in the last 20 years. However, *E. coli* resistant to third-generation cephalosporins, usually producing extended-spectrum beta-lactamases (ESBL), increased from 5.6% in 2014 to 7.4% in 2017. Enterobacteriaceae resistant to fluoroquinolones followed by Enterobacteriaceae resistant to third-generation cephalosporins are probably the most frequently encountered and clinically important antimicrobial-resistant pathogens in Sweden today [8–10].

In septic shock, mortality risk increases if antibiotic treatment is delayed [11, 12]. Early appropriate empirical antibiotic treatment is therefore particularly important in septic shock, and must be initiated without delay before the results of blood cultures are available [13–16]. Since antimicrobial-resistant organisms have become more prevalent in most countries, the choice of appropriate antibiotics becomes increasingly challenging. Accordingly, up-to-date knowledge on the prevalence of microorganisms and their inherent/natural and acquired resistance to antimicrobial agents in serious infection is of major importance if we are to ensure appropriate empiric antimicrobial treatment [14, 17–19]. Furthermore, accurate estimations of AMR are necessary to establish the magnitude of the AMR problem on global, national, regional and local levels [20, 21]. Region Östergötland, with a catchment population of approximately 450,000 inhabitants (5% of the Swedish population), is served by four hospitals and has developed a database cross-linking systems providing microbiological data and mortality data from the patient care administration system. By analyzing data from this registry, we discovered a dramatic increase in community-onset BSI between 2000 and 2013 with comorbidity being the main risk factor for 30-day mortality associated with BSI [3]. The work presented here is a follow-up of a previous population-based study on BSI in the Region of Östergötland, aiming to investigate temporal trends in BSI more thoroughly, including distribution of species, AMR and risk factors for 30-day mortality associated with BSI.

## Material and methods

### Design, setting and population

Setting: The study was carried out in a county in south-east Sweden served by four hospitals: a tertiary care university hospital (600 beds); two general hospitals (310 and 100 beds respectively); and one minor hospital (14 beds). The number of inhabitants in the county increased from approximately 423,000 to 452,000 over the study period, and currently represents approximately 5% of the Swedish population.

Study design: A retrospective cohort study on data from electronic records to describing and analyzing the incidence and 30-day all-cause mortality of culture-confirmed BSI in the Region of Östergötland, Sweden, between January 1, 2008 and December 31, 2016. Data were extracted from the Region Östergötland BSI registry as in a previous study performed in 2000–2013 [3].

### Data collection

The following data were obtained from the Department of Clinical Microbiology in Region Östergötland: date of blood culture; number of aerobic and anaerobic blood culture vials taken; site of puncture; species identification; and susceptibility patterns. The dataset was entered into a secondary database where it was linked to the patient care administration system providing the following data for all patients with blood cultures taken: gender; age; comorbidity; admitting department; date of admission; date of discharge; and mortality. We restricted the dataset to a nine-year period from 2008 through 2016.

### Microbiology

All isolated microorganisms from outpatients care and hospital admissions were analyzed at the species level. Species identification and susceptibility testing were performed at the Region's Clinical Microbiology Department. Matrix-assisted laser desorption ionization time of flight mass spectrometry (MALDI-TOF MS) was used for microbial identification. Clinical susceptibility categories (susceptible, intermediate, and resistant) are defined by cut-offs or breakpoints for different antibiotics and bacteria using the disc diffusion method and Mueller-Hinton agar (MHA). For some bacteria complementary E-test was also applied. Antimicrobial susceptibility clinical breakpoints used during study period were classed as Susceptible (S), Intermediate (I) or Resistant (R) (European Committee on Antimicrobial Susceptibility Testing EUCAST). The intermediate (I) category was excluded from the resistance analyses based on the new definitions of (I) performed by EUCAST [22]. For surveillance purposes EUCAST advice against lumping categories and results should be recorded as S, I and R. If lumping does occur, EUCAST recommends to never lump I+R, only S+I. No major changes in cut-off values for resistance (R), that could have affected susceptibility testing results, were made during the study period.

### Definitions

**Blood cultures.** Blood cultures were taken on clinical indications. One set of blood cultures comprised one aerobic and one anaerobic blood culture bottle. It is recommended that at least two sets of blood cultures are taken simultaneously.

**Positive blood culture.** Defined as the isolation of microorganisms (one or more bacterial or fungal isolates) from a set of blood cultures obtained on the same day in an adult ($\geq$18 yrs). Only initial bacterial or yeast isolates were considered, thus repeat isolates of the same species

with the same antibiogram (or change between S and I or I and R) from the same patient were excluded.

**Non-significant pathogens.** Microorganisms typically belonging to the skin microbiota: (coagulase-negative *staphylococci*, (CoNS); *Micrococcus* spp.; *Bacillus* spp.; *Corynebacterium* spp.; and *Cutibacterium* spp.), were considered probable contaminants and excluded [23].

**Repeat isolate.** Culture of same species with identical resistance pattern isolated during the same admission episode (from admission until hospital discharge). Repeat isolates were excluded.

**Multidrug resistance (MDR).** Non-susceptibility to at least 1 agent in ≥3 antimicrobial classes, (S8 Table) [24, 25].

**Antimicrobial resistance (AMR).** MDR bacteria and MRSA, ESBL-producing Enterobacteriaceae, vancomycin-resistant enterococci not fulfilling the MDR criteria were defined as antimicrobial-resistant (AMR) bacteria.

**BSI episode.** An episode fulfilling the criterion "positive blood culture showing a significant pathogen". If a patient had more than one BSI episode per admission, only the first BSI episode was included in the analyses involving comorbidity and mortality.

**New BSI episode.** Infection caused by a different bacterial or fungal pathogen >3 calendar days after the previous BSI episode or by the same bacterial or fungal pathogen >30 calendar days after the previous BSI episode.

**Defined-daily-dose (DDD).** The DDD is the assumed average maintenance dose per day for a drug used for its main indication in adults according to WHO [26].

**Comorbidity.** Comorbidity based on the International Classification of Diseases (ICD), ICD-10-CA and the updated Charlson Score Index [27–29] (S11 Table).

## Antibiotic consumption

All drug statistics were based on sale statistics measured as defined-daily-doses. Data were collected from national drug consumption statistics published by the Swedish e-Health Agency and accessed through a dedicated secure website [30]. Consumption of systemic antibiotics were analyzed both in outpatient care and hospital (defined either as J01 in the ATC-code for international comparison, or J01, excluding J01XX05 methenamine as the national Swedish quality indicator) was measured as defined-daily-doses (DDD) per 1,000 inhabitants and day. Use of systemic antibiotics in hospital was measured as DDD using hospital admission or hospital days as denominator [30, 31].

## Statistical methods

We assessed change in annual incidence of BSI using linear regression, presenting the change in incidence with a 95% confidence interval (CI). The predominant pathogens were defined at genus or species level, and annual trends were tested using linear regression. Chi-square and t-tests were used for univariate analyses to determine risk factors for 30-day all-cause mortality associated with BSI. Multivariable binomial regression analysis was used to calculate Incidence Rate Ratio and 95% confidence interval and to adjust for confounding factors. Patients with negative blood cultures were used as reference group. The following variables were used in the regression model and investigated as possible independent risk factors for mortality associated with BSI: gender; age; comorbidity; and year of diagnosis. The same variables plus BSI caused by AMR bacteria were investigated as risk factors for 30-day all-cause mortality. The annual antibiotic consumption was tested using linear regression. A p-value < 0.05 was considered statistically significant. All statistical analyses were performed with Stata version 15.1.

### Ethical approval

The Regional Ethics Review Board in Linköping, Sweden approved the study. Informed consent was not required. No details of the patients are disclosed and thus patient identity is secure. (Ref.no:2010/160-31).

## Results

### Incidence of BSI, 30-day mortality and microorganisms

A total of 9,268 BSIs fulfilling the inclusion criteria were recorded between January 2008 and December 2016 and subsequently analyzed (Fig 1). 46% (n = 4,242) of the BSI-episodes represented by patients were female and the median age of all patients with a positive blood culture increased from 69 to 70 years (p = 0.01). The most common agents causing BSI were Gram-negative bacteria (50%), Gram-positive bacteria (38%) other bacteria (8%) and *Candida* spp. (4%). The incidence of BSI increased by 66% from 973 in 2008 to 1,610 in 2016 per 100,000 hospital admissions with an average annual increase of 78 BSI episodes per 100,000 hospital admissions during the study period (p<0.01) (S1 Fig, S2 Table and S3 Table). However, the incidence reached a plateau in 2013.The 30-day all-cause mortality rate amongst patients with BSI increased from 162 per 100,000 hospital admissions per year in 2008 to 189 in 2016; an increase of 17% (p<0.01). Over the study period, a total of 1,237 patients died within 30 days after the date when blood cultures were obtained, giving an overall 30-day mortality of 13%. The 30-day all-cause mortality of patients with BSI and two or more comorbidities increased by 98% (p<0.01) (S2 Fig and S9 Table).

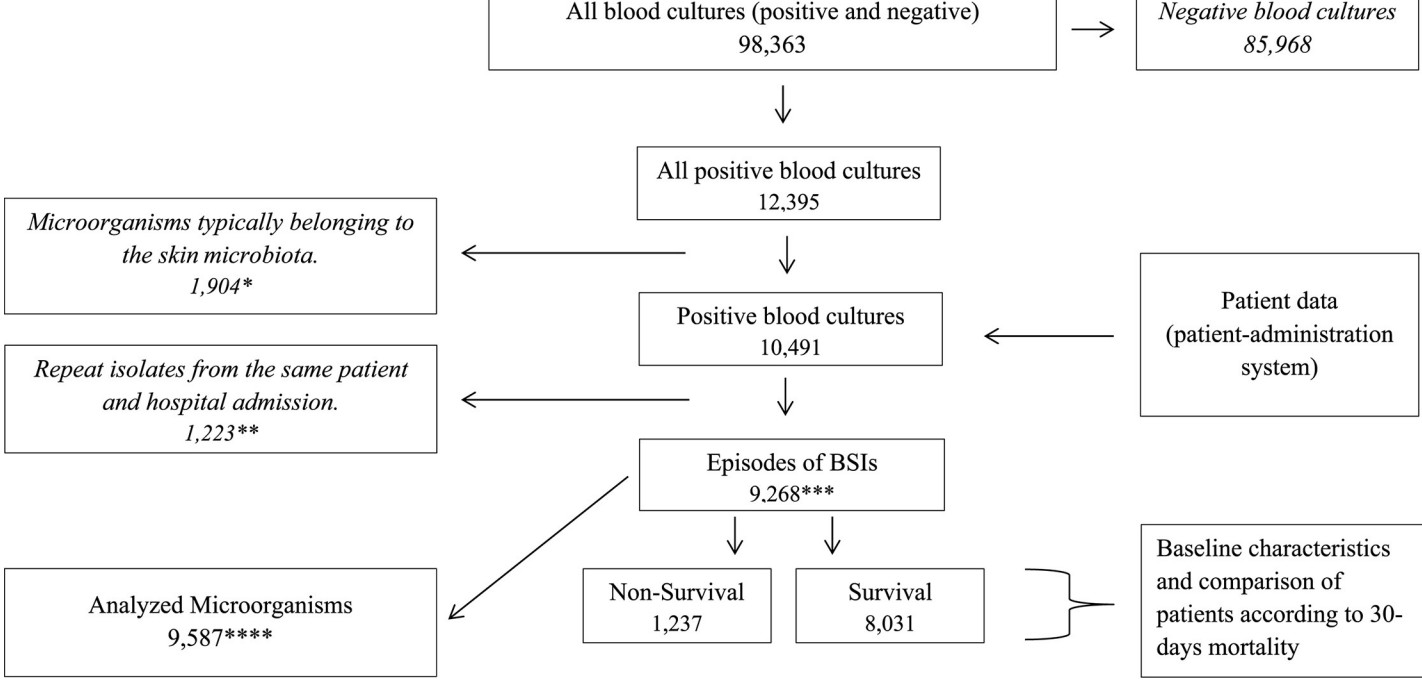

**Fig 1. Flow-chart—study-design.** *Microorganisms typically belonging to the skin microbiota: (coagulase-negative *staphylococci*, (CoNS); *Micrococcus* spp.; *Bacillus* spp.; *Corynebacterium* spp.; and *Cutibacterium* spp.), were considered probable contaminants and excluded **Culture of same species with identical resistance pattern isolated during the same admission episode (from admission until hospital discharge) was excluded. ***9,268 Episodes of BSIs consisted of 8,498 patients (patient with ≥3 BSI-episodes/year = n142, 2 = n317, 1 = n311). ****Total of 9,587 microorganism were analyzed based on BSI-episodes (included polybacterial isolates (n255) and repeat isolate that is not excluded by the definition "repeat isolate" and not cause a new BSI episode (n64).

In all, 9,587 microorganisms were isolated from blood cultures. *Escherichia coli* was the most frequently found cause of BSI, increasing by 93% from 192 to 390 BSIs per 100,000 hospital admissions (p<0.01), with a 30-day all-cause mortality of 8.7%. The second most frequent cause of BSI was, *Staphylococcus aureus* increasing by 55% from 145 to 236 BSIs per 100,000 hospital admissions (p = 0.03) with a 30-day all-cause mortality of 19%. Another species that increased significantly was *Proteus mirabilis* with a 177% increase from 9 to 26 BSIs per 100,000 hospital admissions (p<0.01), and a 30-day all-cause mortality of 13%. Among fungi, a significant increase in *Candida albicans* was recorded, rising by 269% from 11 to 43 BSIs per 100,000 hospital admissions (p = 0.01), with a 30-day all-cause mortality of 29% (S1 Table and S2 Table).

The overall proportion of hospital admissions in which a blood culture was obtained increased from 11 to 19% (p<0.01). The proportion of hospital admissions during which a positive blood culture was found increased from 1.1 to 1.8% (p<0.01), though there was a minor decrease in the proportion of blood cultures that were positive per total number of blood cultures (from 10 to 9%, (p = 0.11)) (S3 Table). Annual distribution of species and associated 30-day mortalities are shown in S2 Table and S4 Table.

## Consumption of antimicrobial agents

During the study period, antibiotics for systemic use dispensed to outpatients decreased by 13% from 11.75 to 10.22 (DDD per 1,000 inhabitants and day (TIND)) (p<0.01) and increased by 17% from 1.28 to 1.50 in hospital wards and polyclinics (DDD/TIND) (p<0.01). The total amount of systemic antibiotics (J01) used in hospital increased by 48% (from 524 to 777 DDD per 1,000 hospital days (2008–2016)) (p<0.01). This corresponds to an increase in DDD per hospital admission of 18% from 2.86 to 3.38. (p = 0.01).When analyzed at the $5^{th}$ ATC level as defined by WHO, different groups of systemic antibiotics used in hospital increased as follows: penicillin combinations including beta-lactamase inhibitors, (J01CR) increased by 224% from 26.4 to 85.5 DDD per 1,000 hospital days (p<0.01); carbapenems (J01DH) increased 53%, 31.3 to 48.0 (DDD per 1,000 hospital days) (p<0.01); beta-lactamase-sensitive penicillins (J01CE) 79%, 41.1 to 73.5 (DDD per 1,000 hospital days) (p<0.01); beta-lactamase-resistant penicillins (J01CF) increased by 109% from 65.8 to 137.8 (DDD per 1,000 hospital days) (p<0.01); penicillins with extended spectrum (J01CA) 48%, 50.3 to 74.4 (DDD per 1,000 hospital days) (p<0.01); and cephalosporins (J01 DB-DE) increased by 24% from 81.2 to 100.9 DDD per 1,000 hospital days (p<0.01) (S5 Table, S6 Table and S7 Table).

## Antimicrobial resistance

Increased resistance to antimicrobial agents was observed during the study period. During the study period we observed significant increases in fluoroquinolone-resistance (3.7 to 7.7% (p = 0.01) and cephalosporin-resistance (2.5 to 5.2% (p = 0.03) amongst Enterobacteriaceae (S10 Table). Theresistance of *E.coli* to ciprofloxacin increased from 9 BSI episodes to 43 (6.7–11%), 2008–2016 (p = 0.02). Furthermore, *E. coli* resistance to tobramycin increased from 2 to 19 (1.0–4.9%) (p = 0.03). Significantly increased piperacillin-tazobactam resistance rates were observed among several Gram-negative bacteria; *Pseudomonas aeruginosa* 0 to 5 (0–19.2%), *Klebsiella oxytoca* 0–3 (0–13%), and *Enterobacter cloacae* 0–6 (0–17.1%) (<0.01). Furthermore, there was an increase in resistance to trimethoprim-sulfamethoxazole 1–7 (11.1–26.9%), (p = 0.03) and ciprofloxacin 0–1 (0–3.8%), (p = 0.03) amongst isolates of *Proteus mirabilis* (S4 Table).

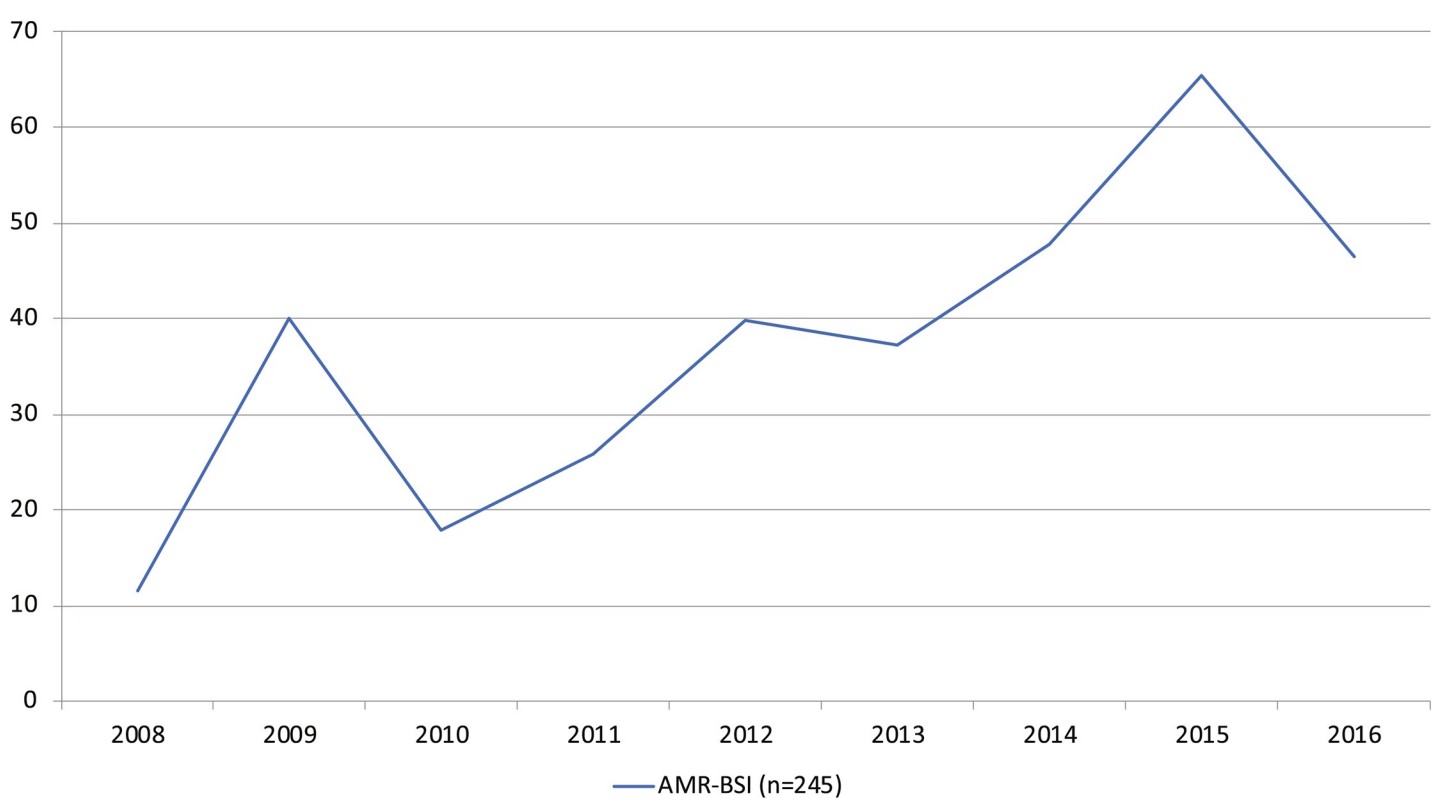

**Fig 2. AMR BSIs per 100,000 hospital admissions and year, 2008–2016.** Incidence of AMR BSI increased by 300% from 12 to 47 per 100,000 hospital admissions, 2008–2016, (Linear regression) (p = 0.01).

### Antimicrobial resistance and 30-day all-cause mortality

A total of 245 AMR BSIs (2.6%) were observed and among those 185 were caused by MDR bacteria; ESBL *E. coli* (ESBL-E) (n = 74), ESBL *Klebsiella pneumoniae* (ESBL-K) (n = 22), MRSA (n = 6), non-ESBL *E. coli* (n = 62), others (n = 38) and non-MDR bacteria (n = 60); (ESBL-E (n = 48), ESBL-K (n = 3), MRSA (n = 7) and VRE (n = 2)). AMR BSIs increased by 300%, from 12 to 47 per 100,000 hospital admissions, 2008–2016 (p = 0.01) (Fig 2 and Table 1). The 30-day all-cause mortality due to BSIs caused by AMR BSI was 9.4%, ESBL-E 6.8%, MDR *E. coli* 4.8% which was lower than the 8.7% 30-day all-cause mortality rate for all *E. coli* BSIs Table 2.

### Univariate and multivariable risk factor analyses

Age, gender, myocardial infarction, congestive heart failure, peripheral vascular disease, cerebrovascular disease, dementia, connective tissue disease (rheumatic disease), renal disease and cancer were significantly associated with 30-day all-cause mortality in univariate analyses. Non-survivors (30-day all-cause mortality due to BSI) had a significantly higher mean Charlson comorbidity score (mean = 3.81, SD = 3.72–3.90 compared to survivors (mean 1.79 (1.73–1.85), p<0.01) Table 3.

Multivariable analyses showed the following to be risk factors for 30-day mortality: age (RR 1.02 (95% CI: 1.02–1.03); one comorbidity RR 2.06 (95% CI: 1.60–2.52); two comorbidities RR 2.79 (95% CI: 2.27–3.42); and three or more comorbidities RR 2.82 (95% CI: 2.31–3.45). Patients with an AMR BSI had a significantly lower 30-day all-cause mortality; RR 0.89 (95% CI: 0.81–0.97) p = 0.01 Table 4.

**Table 1. Distribution of antimicrobial resistant (AMR) bacteria per 100,000 hospital admissions.**

| | | 2008 | 2009 | 2010 | 2011 | 2012 | 2013 | 2014 | 2015 | 2016 | Total | Change %* | Regression per 100,000 hospital admissions** | | | 30-day mortality | |
| --- | --- | --- | --- | --- | --- | --- | --- | --- | --- | --- | --- | --- | AAI** | 95% CI | p-value | Non-survival | Survival |
| **AMR BSI** | | 8 | 29 | 13 | 19 | 30 | 28 | 36 | 48 | 34 | 245 | 305% | 5 | 1.53–8.04 | 0.01 | 23 (9%) | 222 (91%) |
| **MDR-bacteria** | | 3 | 20 | 10 | 18 | 26 | 23 | 23 | 34 | 28 | 185 | . | 4 | 1.62–6.11 | <0.01 | 17 (9%) | 168 (91%) |
| | Escherichia coli | 2 | 6 | 4 | 5 | 9 | 9 | 6 | 11 | 10 | 62 | . | 1 | 0.49–1.66 | <0.01 | 3 (5%) | 59 (95%) |
| | ESBL *E.coli* | 0 | 7 | 3 | 4 | 9 | 9 | 11 | 18 | 13 | 74 | . | 2 | 1.22–3.56 | <0.01 | 5 (7%) | 69 (93%) |
| | ESBL *Klebsiella spp* | 0 | 2 | 3 | 5 | 7 | 1 | 3 | 0 | 1 | 22 | . | 0 | -1.16–0.87 | 0.75 | 5 (23%) | 17 (77%) |
| | MRSA | 1 | 2 | 0 | 1 | 0 | 0 | 2 | 0 | 0 | 6 | . | 0 | -0.52–0.19 | 0.30 | 2 (33%) | 4 (67%) |
| | *Acinetobacter baumannii* | 0 | 0 | 0 | 0 | 0 | 1 | 1 | 0 | 1 | 3 | . | 0 | -0.01–0.33 | 0.06 | 1 (33%) | 2 (67%) |
| | *Pseudomonas aeruginosa* | 0 | 0 | 0 | 1 | 0 | 0 | 0 | 0 | 0 | 1 | . | 0 | -0.17–0.12 | 0.73 | 0 | 1 (100%) |
| | *Proteus mirabilis* | 0 | 0 | 0 | 0 | 0 | 0 | 0 | 1 | 0 | 1 | . | 0 | -0.07–0.20 | 0.27 | 0 | 1 (100%) |
| | *Staphylococcus aureus* | 0 | 2 | 0 | 0 | 0 | 1 | 0 | 0 | 1 | 4 | . | 0 | -0.35–0.30 | 0.86 | 0 | 4 (100% |
| | *Streptococcus pneumoniae* | 0 | 1 | 0 | 2 | 1 | 2 | 0 | 4 | 2 | 12 | . | 0 | -0.01–0.90 | 0.10 | 1 (8%) | 11 (92%) |
| **Non-MDR bacteria** | | 5 | 9 | 3 | 1 | 4 | 5 | 13 | 14 | 6 | 60 | . | 1 | -0.84–2.69 | 0.26 | 6 (10%) | 54 (90%) |
| | ESBL *E.coli* | 4 | 8 | 2 | 1 | 3 | 3 | 11 | 13 | 3 | 48 | . | 1 | -1.13–2.45 | 0.41 | 2 (4%) | 46 (96%) |
| | ESBL *Klebsiella spp* | 0 | 1 | 0 | 0 | 0 | 2 | 0 | 0 | 0 | 3 | . | 0 | -0.33–0.28 | 0.86 | 2 (67%) | 1 (33%) |
| | MRSA | 1 | 0 | 1 | 0 | 0 | 0 | 2 | 0 | 3 | 7 | . | 0 | -0.22–0.66 | 0.28 | 2 (29%) | 5 (71%) |
| | VRE | 0 | 0 | 0 | 0 | 1 | 0 | 0 | 1 | 0 | 2 | . | 0 | -0.12–0.25 | 0.41 | 0 | 2 (100%) |

* Change in rate from 2008–2016 per 100,000 hospital admissions.

**Average annual increase (AAI). Increased microorganism per year per 100,000 hospital admissions (average annual increase), Linear regression.

## Discussion

In this study BSI incidence, 30-day all-cause mortality and AMR-bacteria increased over the study period. Surprisingly, the study showed lower 30-day all-cause mortality among patients with BSI caused by AMR bacteria including ESBL, MRSA and VRE without MDR, compared

**Table 2. Characteristics of all BSIs, AMR BSIs, susceptible *E. coli* BSIs, ESBL *E.coli* and MDR *E. coli* BSIs.**

| | All BSIs | AMR BSIs | Susceptible *E.coli* | ESBL *E.coli* | MDR *E.coli* |
| --- | --- | --- | --- | --- | --- |
| **All** | **9268** | **245** | **3143** | **122** | **62** |
| Male n (%) | 5005 (54) | 78 (62) | 1446 (46) | 79 (64) | 50 (63) |
| Mean Age (SD) | 70 (17) | 67 (17) | 72 (16) | 64 (19) | 66 (18) |
| Charlson Index (SD) | 2.5 (2.7) | 1.8 (2.5) | 2.2 (2.3) | 1.7 (2.8) | 1.7 (2.4) |
| Mortality (%) | 13 | 9.4 | 8.7 | 6.8 | 4.8 |

**Table 3. Baseline characteristics and comparison of patients according to 30-day all-cause mortality.**

|  |  | Non-survivors | Survivors |  |
|---|---|---|---|---|
|  | BSI episodes | n = 1237 (13%) | n = 8031 (87%) | p-value* |
| **Demographics** |  |  |  |  |
|  | **Male** | 692 (56) | 4334 (54) | 0.19 |
|  | Mean age± (SD) | 76±12.8 | 69±18.6 | 0.01** |
|  | >65 years | 79 | 77 |  |
| **Comorbidity** |  |  |  |  |
|  | Myocardial infarction | 223 (18) | 1044 (13) | <0.01 |
|  | Congestive heart failure | 383 (31) | 1526 (19) | <0.01 |
|  | Peripheral vascular disease | 111 (9.0) | 484 (6.0) | <0.01 |
|  | Cerebrovascular disease | 210 (17) | 1044 (13) | <0.01 |
|  | Dementia | 73 (5.9) | 322 (4.0) | 0.01 |
|  | Chronic pulmonary disease | 198 (16) | 1124 (14) | 0.12 |
|  | Connective Tissue Disease-Rheumatic Disease | 99 (8.0) | 482 (6.0) | 0.03 |
|  | Peptic ulcer disease | 49 (4.0) | 242 (3.0) | 0.14 |
|  | Mild liver disease | 50 (4.0) | 401 (5.0) | 0.89 |
|  | Diabetes without chronic complications | 309 (25) | 1767 (22) | 0.05 |
|  | Diabetes with chronic complications | 76 (6.1) | 403 (5.0) | 0.21 |
|  | Paraplegia and hemiplegia | 72 (5.8) | 482 (6.0) | 0.99 |
|  | Renal disease | 210 (17) | 884 (11) | <0.01 |
|  | Cancer | 445 (36) | 2010 (25) | <0.01 |
|  | Moderate or severe liver disease | 61 (4.9) | 162 (2.0) | <0.01 |
|  | Metastatic carcinoma | 173 (14) | 643 (8.0) | <0.01 |
|  | HIV/AIDS | 0 (0) | 7 (0.1) | 0.99 |
| **Number of comorbidities** |  |  |  |  |
|  | 0 | 147 (12) | 2425 (30) | <0.01 |
|  | 1 | 307 (25) | 2101 (26) | 0.36 |
|  | 2 | 302 (24) | 1453 (18) | <0.00 |
|  | >2 | 481 (39) | 2021 (25) | <0.01 |
| **Charlson comorbidity score** |  | 3.81 (3.72–3.90) | 1.79 (1.73–1.85) | <0.01** |

*Chi²-analysis

** Student-T-test

**Table 4. Multivariable analyses–factors for 30-day all-cause mortality in BSI.**

|  | BSI Incidence |  |  | 30-day mortality |  |  |
|---|---|---|---|---|---|---|
|  | Risk Ratio* | 95% CI | p-value | Risk Ratio* | 95% CI | p-value |
| Age | 1.01 | (1.01–1.01) | <0.01 | 1.02 | (1.02–1.03) | <0.01 |
| Male | 0.99 | (0.95–1.03) | 0.58 | 1.05 | (0.94–1.17) | 0.39 |
| **Number of comorbidities** |  |  |  |  |  |  |
| 0 | 1 |  |  | 1 |  |  |
| 1 | 1.25 | (1.18–1.32) | <0.01 | 2.06 | (1.68–2.52) | <0.01 |
| 2 | 1.32 | (1.24–140) | <0.01 | 2.79 | (2.27–3.42) | <0.01 |
| ≥3 | 1.58 | (1.50–1.67) | <0.01 | 2.82 | (2.31–3.45) | <0.01 |
| AMR BSI** |  |  |  | 0.89 | (0.81–0.97) | 0.01 |

* Multivariate binomial regression analysis

** MDR bacteria and MRSA, ESBL, VRE)

to those with susceptible bacteria. The dominant risk factor for 30-day all-cause mortality associated with BSI was comorbidity, which agrees with previous studies [3, 32, 33]. The incidence of BSI and associated 30-day all-cause mortality per 100,000 hospital admissions increased over the study period by 66%, and 17% respectively which is consistent with the results of other studies [4, 34, 35]. The incidence of BSI and the 30-day all-cause mortality reached a plateau in 2013 and thereafter mortality decreased between 2013 and 2016. Other studies have reported a stable or even decreased incidence of BSI and associated mortality [36, 37]. The overall 30-day all-cause mortality among patients with BSI was 13%, which is similar to that estimated in Europe and North America, and also seen in Finland [2, 4]. There are several factors that could explain the increase in the annual incidence of BSI up to 2013, including the increase in the number of blood cultures taken per hospital admission; a direct result of increased compliance with the principle of taking blood cultures before starting antibiotic treatment. However, the proportion of positive blood cultures did not change significantly over the study period. If potential areas for improvement are to be found, other possible explanations for the observed increase in BSI must be studied, both in previously healthy patients and those with comorbidities.

In the present study, *E. coli* was the most common cause of bloodstream infection; the incidence increasing by almost 100% between 2008 and 2016. We also found an increase in BSI caused by *E. coli* resistant to quinolones, cephalosporins and aminoglycosides, which concurs with global trends [34, 37–40]. BSI caused by ESBL-producing Enterobacteriaceae increased during the study period, though the rate was low compared to many other European countries; the numbers, however, are increasing [7]. The reason for the rapid dissemination of ESBL-producing Enterobacteriaceae is likely multifactorial with travel and migration as driving forces [41, 42].

Antibiotic hospital consumption increased by approximately 50% measured as DDD per 1,000 hospital days (p<0.01) comprising both narrow- and broad-spectrum drugs (S6 Table). Several factors could explain this. First of all, BSIs per 100,000 hospital days increased by 108% (S3 Table) and per 100,000 hospital admissions by 66%; hence the increased use of antibiotics in hospitals. Second, the number of patients with multiple comorbidities increased, and since these patients are at greater risk for severe illness they were probably prescribed more antibiotics [43]. Third, modern guidelines recommend higher and more frequent doses of antibiotics which would naturally lead to increased consumption as measured by DDD based on standard doses for the main indication [44–49]. This study was not designed to see if the decrease in antibiotic treatment in outpatients correlated with the increase in antibiotic use in hospital, since we did not consider antibiotic use in individual patients.

Similar to trends in the other Nordic countries, the use of piperacillin-tazobactam (PTZ) and amoxicillin-clavulanic acid increased rapidly [50]. It is interesting to note a concurrent increase in PTZ resistance among *Klebsiella oxytoca* and *Enterobacter cloacae* as well as *Pseudomonas aeruginosa*, but the study was not designed to show causal relationship between consumption and emergence of resistance to PTZ. In other studies, however, degree of exposure to PTZ has paralleled the emergence of PTZ resistance among *P. aeruginosa* when cephalosporins have been replaced by PTZ [51]. This warrants further investigation since *P. aeruginosa* can cause healthcare-associated infections that are difficult to treat.

Only 2.6% of all BSIs were caused by AMR bacteria. The restricted use of systemic antibiotics in the Swedish primary healthcare system and in animals, probably explains why we have a relatively low prevalence of AMR bacteria compared to other European countries. In the hospital setting however, antibiotic consumption is similar to other European countries [52, 53]. Other concomitant factors that could explain the low level of AMR rates in Sweden include a high degree of food safety, improved hygiene and infection prevention measures, and

meticulous sanitation. In the design of this study we decided to use the generally accepted definition of multidrug resistance (MDR) *i.e.* non-susceptibility to at least 1 agent in ≥3 antimicrobial classes [24, 25]. However, application of this MDR definition to our data would exclude a significant number of MRSA, ESBL-producing Enterobacteriaceae not fulfilling the MDR criteria thereby underestimating the frequency of AMR. Thus, we report MDR but also AMR without multidrug resistance (MRSA, ESBL-producing Enterobacteriaceae, vancomycin-resistant enterococci not fulfilling the MDR criteria.

Surprisingly, multivariable analyses showed a lower 30-day all-cause mortality among patients with BSI caused by AMR bacteria including ESBL, MRSA and VRE without MDR, compared to those with susceptible bacteria [54], while other studies have reported the opposite [32, 55, 56]. Further studies are needed to explain the lower mortality rate found in this group. In the present study, patients with a BSI caused by AMR bacteria were younger and had less comorbidity. It is possible that people of this age are more exposed to AMR-bacteria because of frequent travel. Furthermore, patients with AMR BSI may receive longer intravenous antibiotic therapy and spend more time in hospital as well, thereby may reducing the chance of recurrent infection. A current lack of resources obliges us to reduce in-hospital times with the result that patients with sensitive bacteria might discharged prematurely. In Sweden, patients with AMR bacteria are usually treated by an infectious disease specialist. Infectious disease consultation has been shown to improve outcome in S. aureus sepsis [57] and this may also have influenced the treatment of BSI caused by AMR bacteria of other species including ESBL-E. coli which was the most prevalent AMR bacteria found in this study. Improved care due to involvement of an ID specialist may be an explanation for the better outcome and lower mortality rate among patients with AMR bacteria. [54, 58–60]. This warrants further study and we are planning a case-control study with extensive data on patients and bacteria, including virulence factors, in order to gain a better insight into why patients with ESBL-E. coli BSI have a better outcome in our setting.

A limitation of this study was that we did not divide the cohort into community-acquired and hospital-acquired infections due to the difficulty in defining these groups. Another limitation was that risk factor and mortality analysis was performed on the first BSI even if a patient admitted for a community-acquired BSI suffered a hospital-acquired BSI during the same admission. However, since only 2% of patients had more than one BSI during the same admission, this limitation could only have a minor influence on the results. Furthermore, CoNS was considered a contaminant, thus excluding central line-associated BSIs (CLABSI) caused by CoNS; the reason being that these analyses are complicated, and our aim was to simplify things by using a bacteremia database as a tool in our effort to improve the management of BSI. A major limitation is that individual patient data on severity of disease, site of infection and appropriate antibiotic treatment were not available in our database; we were thus unable to assess any association between antibiotic use and risk for antibiotic-resistant infection, or between specific empirical regimens and outcome. Nor could we determine if delay in appropriate antibiotic treatment was a risk factor for mortality as shown by Andersson et al [12] in a similar setting. Furthermore, other causes of mortality such as myocardial infarction, pulmonary embolus or respiratory failure were only registered as comorbidity and not evaluated as a primary cause of death. Since this was mainly an explorative analysis, we have not adjusted the p-values; a measure usually taken when testing multiple hypotheses.

## Conclusion

Comorbidity was the predominant risk factor for 30-day all-cause mortality associated with BSI in this study. The burden of AMR was low and not associated with increased mortality.

Patients with BSIs caused by AMR bacteria (MDR, MRSA, ESBL and VRE) were younger, had fewer comorbidities, and the 30-day all-cause mortality was lower in this group. The reason for this will be the subject of further studies.

## Supporting information

**S1 Fig. Bloodstream infections per 100 000 hospital admissions and year.**
(PDF)

**S2 Fig. 30-day all-cause mortality due to BSI per 100 000 hospital admissions and year.**
(PDF)

**S1 Table. Increase in BSI per microorganism per 100,000 hospital admissions and year, 2008–2016.**
(PDF)

**S2 Table. Distribution of most commonly occurring microorganisms causing BSI, 2008–2016.**
(PDF)

**S3 Table. Blood culture characteristics (hospital admission.** blood cultures. microorganism. blood culture per hospital admission. and positive blood culture per total number of blood cultures and hospital admissions).
(PDF)

**S4 Table. Incidence of BSI per microorganism and antimicrobial resistance (2008–2016).**
(PDF)

**S5 Table. Antibacterials for systemic use (J01) excluding metenamine (J01-J01XX05) measured as defined-daily-doses (DDD) per 1.000 inhabitants and day (TIND).**
(PDF)

**S6 Table. Amount of antibacterials for systemic use (J01) used on hospital wards and polyclinics measured in defined-daily-doses (DDD) per 1,000 hospital days.**
(PDF)

**S7 Table. Amount of antibacterials for systemic use (J01) used on hospital wards and polyclinics measured in defined-daily-doses (DDD) per hospital admission.**
(PDF)

**S8 Table. Categories and agents used to define MDR (worksheet for categorizing isolates).**
(PDF)

**S9 Table. Comorbidity per 100,000 hospital admissions and year (overall BSIs and 30-day all-cause mortality).**
(PDF)

**S10 Table. Antibiotic resistance in Enterobacteriaceae (2008–2016).**
(PDF)

**S11 Table. The Charlson Comorbidity Index (Updated Weight),**
(PDF)

## Author Contributions

**Data curation:** Martin Holmbom, Lennart E. Nilsson, Mats Fredrikson, Håkan Hanberger.

**Formal analysis:** Martin Holmbom, Mats Fredrikson, Håkan Hanberger.

**Investigation:** Martin Holmbom.

**Methodology:** Martin Holmbom, Lennart E. Nilsson, Mats Fredrikson, Håkan Hanberger, Åse Östholm Balkhed.

**Project administration:** Håkan Hanberger, Åse Östholm Balkhed.

**Resources:** Håkan Hanberger.

**Software:** Martin Holmbom, Mats Fredrikson.

**Supervision:** Lennart E. Nilsson, Christian G. Giske, Mats Fredrikson, Håkan Hanberger.

**Validation:** Martin Holmbom, Lennart E. Nilsson, Christian G. Giske, Mats Fredrikson, Håkan Hanberger, Åse Östholm Balkhed.

**Writing – original draft:** Martin Holmbom, Vidar Möller, Lennart E. Nilsson, Christian G. Giske, Mamun-Ur Rashid, Mats Fredrikson, Anita Hällgren, Håkan Hanberger, Åse Östholm Balkhed.

**Writing – review & editing:** Martin Holmbom, Christian G. Giske, Mats Fredrikson, Håkan Hanberger, Åse Östholm Balkhed.

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
