## [Decision Letter · Decision Letter 0]

19 Dec 2019

PONE-D-19-29217

Low incidence of antibiotic-resistant bacteria in south-east Sweden: an epidemiologic study on 9268 cases of bloodstream infection.

PLOS ONE

Dear Prof Hanberger, 

Thank you for submitting your manuscript to PLOS ONE. After careful consideration, we feel that it has merit but does not fully meet PLOS ONE’s publication criteria as it currently stands. Therefore, we invite you to submit a revised version of the manuscript that addresses the points raised during the review process.

The reviewers presented several comments, see below. Please address their comments and revise accordingly. The reviewers were not in agreement concerning the manuscript, with the main concern raised being missing variables important for the analysis. I am aware that these variables would not be possible to complete from your database, however this issue should be clearly stated in the limitations section.

We would appreciate receiving your revised manuscript by 22-Jan-2020. To enhance the reproducibility of your results, we recommend that if applicable you deposit your laboratory protocols in protocols.io, where a protocol can be assigned its own identifier (DOI) such that it can be cited independently in the future. For instructions see: http://journals.plos.org/plosone/s/submission-guidelines#loc-laboratory-protocols

We look forward to receiving your revised manuscript.

Kind regards,

Dafna Yahav

Academic Editor

PLOS ONE

Journal Requirements:

2. We ask that you please include in your methods section, a statement that the need for informed consent was waived by your ethics committee. Thank you for including this information in your on-line ethics statement.

4. We note you have included a table to which you do not refer in the text of your manuscript. Please ensure that you refer to Table 1 in your text; if accepted, production will need this reference to link the reader to the Table.

Reviewers' comments:

Reviewer's Responses to Questions

**Comments to the Author**

1. Is the manuscript technically sound, and do the data support the conclusions?

Reviewer #1: Partly

Reviewer #2: Partly

Reviewer #3: Yes

Reviewer #4: Partly

Reviewer #5: Partly

2. Has the statistical analysis been performed appropriately and rigorously? 

Reviewer #1: Yes

Reviewer #2: I Don't Know

Reviewer #3: Yes

Reviewer #4: Yes

Reviewer #5: Yes

3. Have the authors made all data underlying the findings in their manuscript fully available?

Reviewer #1: Yes

Reviewer #2: No

Reviewer #3: Yes

Reviewer #4: Yes

Reviewer #5: Yes

4. Is the manuscript presented in an intelligible fashion and written in standard English?

Reviewer #1: Yes

Reviewer #2: Yes

Reviewer #3: Yes

Reviewer #4: Yes

Reviewer #5: Yes

5. Review Comments to the Author

Reviewer #1: General comment

This is a retrospective cohort study analysing blood culture isolates form a Swedish county, covering the years 2008-2016. The authors demonstrate that BSI have been increasing over time, that BSI caused by resistant pathogens are rare in this specific setting and that 30-day mortality is mainly related to the patient comorbidities. The main limitation is the lack of important confounder variables in the mortality analysis (e.g. appropriate antibiotic treatment, site of infection, severity of illness). The discussion needs restructuring.

Detailed comments

Introduction

- line 78: Write out BSI when first mentioning

Methods

- line 118: are isolates from outpatients outside the four hospitals also represented in the database?

- line 195: you are performing dozens of statistical tests in your analysis. Consider adjusting the p-values for multiple testing, or mention this as a limitation.

Results

- line 219: 9268 BSI from how many patients?

- line 231: regarding increased 30d mortality over time: Could it be that follow-up was different over time? In other words, how sure can you be about those patients for whom no death was reported? Please comment.

- line 254: what is the difference between the 17% increase mentioned on line 253 and the 48% on line 254?

- line 256: what does “analysed at the 5th ATC level” mean?

- Table 1: Comment: From an epidemiological perspective, separation between MDR and non-MDR ESBL does not make much sense.

- Table 3: p-value for gender is missing

- Table 4: a limitation of your analysis is the small number of potential confounders. What about severity of disease? Site of infection? Lower mortality with AMR is strange and is probably a result of residual confounding.

Discussion

- Paragraph 1 is not well structured. You are intermingling all aspects of your results in the first paragraph, which is puzzling to the reader. This should be a summary of your main findings. In general, do not repeat details of your results in the discussion and do not refer to Tables if not necessary.

- line 361: I am not sure. At the same time guidelines recommend shorter courses of antibiotic treatment for many indications compared to 10 years ago. If you stick to this argument, please give a reference.

Reviewer #2: Methods

1. Were all BSIs diagnosed in-hospital or in the community as well? Please specify

2. What methods do you use for pathogen identification and antimicrobial susceptibility?

3. You defined that the intermediate (I) category was excluded from the resistance analyses. If the numbers are small it may be negligible, but consider adding it to the resistant (R) category.

4. In analyses involving mortality only the first BSI per admission was included. How many patients had recurrent admissions with BSIs? If a significant amount then that introduces bias.

5. From the results I understand that antibiotic consumption data was based on dispensing. Please specify in methods.

Results

1. Add the excluded BSIs and reasons for exclusion.

2. Data are lacking regarding source of infection, place of acquisition, number of patients (not only number of BSIs)- if available please add

3. Antimicrobial resistance- instead of stressing specific bacteria, e.g. Klebsiella oxytoca resistant to PTZ (only 0-3 cases), I would specify general data- quinolone-resistant Enterobacteriaceae, PTZ-resistant Enterobacteriaceae, etc.

4. Table 2- I would change "univariate analysis" to "characteristics"

5. Table 3- specifying male and female both is redundant, there are no units for some of the categories (>65- %, charlson score- maybe median (IQR)), for mean (age) add SD.

6. Table 4- this is the only place that refers to the multivariate analysis for BSI incidence. There is no reference to these results in the text. Also this means that data on patients without BSI had to be extracted and if so, however there is no mention of these patients?

Discussion

1. The result that AMR BSI confers protection against mortality is, as mentioned, very surprising. Explaining this by the observation that they were younger and had less comorbidity is not enough as this is surprising as well. Also, I don’t understand the claim that they were less subject to treatment limitation

Grammar

1. Line 89- pathogens instead of agents

2. Line 126- in 2000-2013

Reviewer #3: I don't have many comments. I think this was a good piece of research, well-written with appropriate statistics. My only suggestions is that the Definitions section should come in an appendix rather than in the middle of the manuscript and the thought that the reason that the mortality burden of AMR was low was perhaps due to the restricted use of antibiotics in primary care in Sweden. The authors may want to suggest this.

Reviewer #4: The authors performed a nine-year observational retrospective study aimed at describing the epidemiology of BSIs in a county in South-East Sweden. Data on BSI episodes were gathered from the microbiology laboratory database and linked to electronic health records available for the same hospitalization. At the same time the authors describe the trend of antimicrobial consumption in the same hospitals by analyzing DDDs/patient days and consumption in the outpatient setting by analyzing DDs/inhabitants.

The study shows a rising trend in the incidence of BSIs and also a rising 30-day mortality rate in patients diagnosed with bacteremia. In-hospital antimicrobial consumption is also increasing, probably as the consequence of the increasing age and comorbidities of admitted patients. The authors already published part of this dataset, showing the relationship between comorbidities and mortality in bacteremic patients. In this new study, they analyzed a larger cohort adding data on antimicrobial resistance and antimicrobial consumption.

My main concern with this study is the unclear study design. When dealing with large databases, it is very common to find significant results and infer causal relationship from these results. However, this approach should generally be avoided if there is no a pre-defined/biologically plausible hypothesis grounding the analysis. In this case, for example, the authors found a significant association between comorbidities and 30-day mortality, proving a biologically plausible hypothesis (already present in previous results from the same cohort). However, when coming to AMR, the analysis shows that AMR has a significant protective effect, even after correcting with comorbidities and age. Unfortunately, with the available data this cannot be explained and proves the difficulty of dealing with this type of data. When testing risk factors for mortality in BSIs we should probably not accept analyses that cannot include relevant patient-level data, such as antibiotic therapy and appropriate measures of infection severity.

Few other comment on the text follows:

Abstract: the study-design should be mentioned in the abstract and in the methods it should be clearly stated how the data collection was performed and how the study was planned (retrospective cohort from electronic records..).

AMR/MDR definition: definition used is quite tricky because MRSA should be always considered MDR. It is difficult to imagine that approximately 1/3 of ESBLs are resistant to less than 3 classes (so not falling under the MDR definition). I would suggest reconsidering this categorization dividing pathogens according to their phenotypic resistance, rather than number of classes (FQ-R E. coli; ESBL-Ent; MRSA, VRE…).

Discussion:

-Increasing incidence and blood cultures: in the discussion the authors state that the increase number of BSIs is probably not due to the growing number of patients tested because the proportion of positive blood cultures remain the same. I would advise to report somewhere the overall number of blood cultures performed, rather than the proportion to better exclude this confounder.

-Line 379: association between AMR and antimicrobial consumption, sanitation, food safety is a very complex issue. I would advise rephrasing as ‘…these are all concomitant factors that could explain the low level of AMR rates in Sweden’.

-Line 386: what do the author mean with treatment limitation? The association with AMR remains significant even if adjusting per age and comorbidities, so they probably should not be considered as relevant factors.

Reviewer #5: This study investigated the risk factors for 30-day mortality among hospitalized bacteraemia patients in a county in Sweden over a nine-year period, using routinely-collected linked data and taking into consideration antibiotic susceptibility. Overall, I read this article with interest and appreciate how much work the authors have done and how much data they have provided. I do, however, think amendments and additional clarification need to be provided before this article can be considered for publication.

Main amendments:

Line 147 – Why were the intermediate isolates excluded? In many other studies “intermediate” and “resistant” have been combined to the category of “non-susceptible” and compared with “susceptible”. It could be a useful sensitivity analysis to investigate whether including these isolates in the “non-susceptible” category alters the results significantly. Alternatively, quantifying how many intermediate isolates were excluded and what the rationale behind this decision was would be helpful.

Lines 159-161 and lines 166-168 – How did the authors handle non-identical antibiograms? Was the most resistant antibiogram chosen (standard practice)? What about polymicrobial blood cultures?

Lines 171-173 – Why was AMR not simply defined as any culture resistant to any of the antibiotics included in the standard panel? Using the definition outlined by the authors, were some antibiotic-resistance profiles missed?

Lines 174-179 – How was the new BSI episode definition devised (references, expert opinion etc.)? What is the rationale behind only using the first BSI episode within a hospitalization? I would imagine that patients with more than one BSI episode (defined as being caused by different aetiogical agents) would be at a higher risk of dying within 30 days?

Lines 182-183 – Outlining the conditions that make up the Charlson Comorbidity Index would be useful, as well as how the Index is calculated and whether the CCI was the only measure of comorbidity used or whether comorbidities were looked at separately.

Definitions – Please outline whether mortality was 30-day all-cause mortality following a BSI episode and/or how mortality was attributed to the BSI. If the measure was 30-day all-cause mortality following a BSI, please update the wording of this throughout the manuscript.

Lines 234-242 – It would possibly be more informative to report on the relative proportions of different bacterial species causing BSI between 2008 and 2016 as opposed to the increase for each pathogen. Reporting pathogen increase could simply be related to an overall increase in BSI incidence, whereas reporting proportions of aetiological agents could give an indication of changes in the most prevalent species/pathogenicity etc.

Table 3 – Why are there two measures of age? Why is comorbidity measured in three separate ways (individual conditions, number of comorbidities and CCI)? How do the authors know that the individual comorbidities were not the cause of death?

Table 4 – Why was number of comorbidities chosen to be part of the adjusted model instead of CCI? CCI showed a significant relationship with the outcome in the univariate analysis and I would argue is a superior measure of overall illness and frailty as it applies different weightings of severity to different conditions to compile an overall composite measure of multi-comorbidity.

Lines 359-361 – Is the proportion of patients with multiple comorbidities increasing over time after adjusting for changes in demographics (age) or is this simply due to an aging population? Perhaps worth discussing a bit more.

Lines 361-362 – Are “modern guidelines” truly recommending “higher and more frequent doses of antibiotics” in Sweden? This is in contradiction with most prescribing guidelines in other high-income settings as antibiotics are recommended for fewer conditions, delayed antibiotic prescribing or “wait and see” approaches are more frequently advised and shorter antibiotic durations are more frequently recommended as first line treatment.

Line 375 – Could the authors please provide a bit more context around the “restricted use of systemic antibiotics” in Sweden? What kinds of policies have been implemented to achieve this?

Lines 382-388 – As this is quite a surprising result which is not in line with previous comparable studies, I believe this requires more discussion as to why this particular patient group (younger, less ill patients) had a higher rate of MDR infections, leading to the effect on morbidity. Could there be factors related to travel? Better antibiotic stewardship in long-term care facilities and primary care making the elderly less likely to receive antibiotics (although this seems unlikely, but perhaps possible in Sweden)? More discussion around possible contributing factors would be welcome.

Lines 404-406 – This is indeed a major limitation, the possible implications of which should be discussed and explored further.

Smaller amendments:

Line 78 – Define BSI in the first instance

Line 84 – “…has not increased significantly in the last…”

Line 86 – Is this increase in Sweden in 2017?

Line 88 – Please avoid the use of “probably”

Line 94 – “Since antimicrobial-resistant organisms have become…”

Line 108 – Other information systems such as? Please specify

Line 123 – I would call this a retrospective cohort study – while the authors do present descriptive statistics, a comparative approach is taken when modelling mortality outcomes.

Study design – It would be helpful to include the patient characteristics defining the cohort (age, gender, any other inclusion criteria besides just the region and the clinical diagnosis)

Lines 192-193 – It is unclear as to whether the systemic antibiotics analysed were both community and hospital or just hospital. Is it possible to differentiate this within the registry so as to accurately define the numerator? Was antibiotic consumption aggregated by the region – please clarify?

Line 220 – Please remove “approximately”

Lines 221-222 – Why not report on the median age over all the years of the study?

Line 223 – Please refer to Table S4 and include percentages in the table.

Line 243 – Is this the overall proportion of hospital admissions due to any cause or just BSI?

Lines 244-247 – This sentence seems to contradict itself, please clarify

Lines 254-255 – This sentence seems to contradict the previous sentence (both reporting hospital prescribing of systemic antibiotics but with different rates), please clarify

Antimicrobial resistance – It seems slightly sporadic which trends the authors chose to highlight - were they the largest changes, the most important drug-bug combinations clinically in the region etc.? Table S6 is unclear with respect to percentages and counts, please consider making this table easier to interpret

Lines 286-288 – This breakdown for 30-mortality is slightly confusing, was ESBL-E not included in AMR BSI? Or was AMR BSI not including ESBL-E or MDR-E? If so, why?

Table 1 – This table is cut off in my PDF and therefore cannot be properly interpreted

Table 2 – This appears to be presenting baseline characteristics, not the results of univariate analyses, please amend

Line 318 – Was age a year-on-year change? If so, I would suggest using a categorical variable for age instead in order to report a more meaningful increase in risk

Line 339 – Only one factor is mentioned, are there others?

Line 401 – “…association between antibiotic use and antibiotic-resistant infection at the patient-level…”

6. PLOS authors have the option to publish the peer review history of their article (what does this mean?). If published, this will include your full peer review and any attached files.

Reviewer #1: No

Reviewer #2: No

Reviewer #3: Yes: Brian Bell

Reviewer #4: Yes: Elena Carrara

Reviewer #5: No

---

## [Author Response · Author response to Decision Letter 0]

29 Feb 2020

Paper PONE-D-19-29217

To the Editor

Thank you for reviewing our manuscript “Low incidence of antibiotic-resistant bacteria in south-east Sweden: an epidemiologic study on 9268 cases of bloodstream infection” and for inviting us to submit a revised version of the manuscript that addresses the points raised in the review process.

We have revised the text according to the referees’ suggestions and below are our responses to the referees’ questions and comments. 

We have also uploaded a 'Revised Manuscript with Track Changes'.

Please let us know if there is anything in our reply that needs further clarification.

We hope our revision is now acceptable for publication.

Yours faithfully

......................................................................................

Håkan Hanberger, MD, Professor

---

## [Editor Report · Decision Letter 1]

3 Mar 2020

Low incidence of antibiotic-resistant bacteria in south-east Sweden: an epidemiologic study on 9268 cases of bloodstream infection.

PONE-D-19-29217R1

Dear Dr. Hanberger,

We are pleased to inform you that your manuscript has been judged scientifically suitable for publication and will be formally accepted for publication once it complies with all outstanding technical requirements.

With kind regards,

Dafna Yahav

Academic Editor

PLOS ONE
---

## [Editor Report · Acceptance letter]

13 Mar 2020

PONE-D-19-29217R1 

Low incidence of antibiotic-resistant bacteria in south-east Sweden: an epidemiologic study on 9268 cases of bloodstream infection. 

Dear Dr. Hanberger:

I am pleased to inform you that your manuscript has been deemed suitable for publication in PLOS ONE. Congratulations! Your manuscript is now with our production department. 

With kind regards,

on behalf of

Dr. Dafna Yahav 

Academic Editor

PLOS ONE